# Profilin 1 and Mitochondria—Partners in the Pathogenesis of Coronary Artery Disease?

**DOI:** 10.3390/ijms22031100

**Published:** 2021-01-22

**Authors:** Elżbieta Paszek, Wojciech Zajdel, Tomasz Rajs, Krzysztof Żmudka, Jacek Legutko, Paweł Kleczyński

**Affiliations:** 1Clinical Department of Interventional Cardiology, John Paul II Hospital, 31-202 Krakow, Poland; elzbieta.m.paszek@gmail.com (E.P.); kwzajdel@gmail.com (W.Z.); tomaszrajs@gmail.com (T.R.); zmudka@icloud.com (K.Ż.); jacek.legutko@uj.edu.pl (J.L.); 2Department of Interventional Cardiology, Institute of Cardiology, Jagiellonian University Medical College, 31-202 Krakow, Poland

**Keywords:** coronary artery disease, mitochondria, profilin 1

## Abstract

Atherosclerosis remains a large health and economic burden. Even though it has been studied for more than a century, its complex pathophysiology has not been elucidated. The relatively well-established contributors include: chronic inflammation in response to oxidized cholesterol, reactive oxygen species-induced damage and apoptosis. Recently, profilin 1, a regulator of actin dynamics emerged as a potential new player in the field. Profilin is abundant in stable atherosclerotic plaques and in thrombi extracted from infarct-related arteries in patients with acute myocardial infarction. The exact role of profilin in atherosclerosis and its complications, as well as its mechanisms of action, remain unknown. Here, we summarize several pathways in which profilin may act through mitochondria in a number of processes implicated in atherosclerosis.

## 1. Introduction

Atherosclerosis is a chronic inflammatory and degenerative disease of large and medium-sized arteries. Its complications are a leading cause of death worldwide [1]. Clinical manifestations of atherosclerosis depend on lesion location, plaque burden and its composition, which translates to stability. The main complications with the biggest toll arise from thromboembolic complications due to plaque rupture or ulceration with subsequent thrombus formation, and again—depending on the vascular bed—include: myocardial infarction, stroke, acute limb ischemia and other embolic complications [2].

Several risk factors for clinically significant atherosclerosis are established, including: male gender, age, hypertension, blood lipid disorders, diabetes mellitus, smoking, chronic kidney disease, lack of physical activity and genetic predispositions [2,3]. So far, causal therapies for atherosclerosis have concentrated on lipid-lowering and anti-inflammatory strategies (involving mostly statins), but they have shown to be less successful than anticipated [4]. Newer, more aggressive anti-inflammatory approaches such as anti-interleukin-1β antibodies have shown to be effective, but at the price of a higher risk of lethal infections [5]. Invasive treatment of coronary artery disease either with percutaneous angioplasty or with coronary artery bypass grafting, although principally lifesaving, is still associated with considerable immediate and long-term risks, depending on the clinical setting [6,7]. These factors underscore the complexity of atherosclerosis and coronary artery disease and justify further investigation into the molecular mechanisms underlying the disease with a potential to develop new therapeutic targets.

Recently mitochondrial pathways have been explored regarding atherosclerosis, yeldieng some exciting results. Profilin 1 is a relatively new and still understudied player in the field. A closer look into several common pathways between profilin and mitochondrial proteins might shed light on the direction of future studies.

## 2. A Molecular Insight into Atherosclerosis

The formation of atherosclerotic plaque begins with an accumulation and oxidation of apolipoprotein B–containing lipoproteins (LDLs) within the artery wall. This constitutes a signal for the activation of the endothelium, as well as an immune response. As the stimulus—oxidized LDL (oxLDL)—acts in a continuous manner, this causes a chronic inflammation within the vessel wall. An activation of the nuclear factor-κB (NF-κB) transcription factor and a switch in gene expression leads to a dramatic change of the endothelial cells’ (ECs) phenotype [8]. The ECs express adhesion molecules: Intercellular Adhesion Molecule 1 (ICAM-1), Vascular Cell Adhesion Molecule 1 (VCAM-1), selectin-E, selectin-P, chemokines attracting monocytes (Monocyte Chemoattractant Protein-1, Interleukin-8) and T lymphocytes (CXC) [9]. The activated monocytes migrate into the vessel wall. ECs express the von Willebrand factor, which binds glycoprotein Ib on platelets, leading to their activation. Activated platelets, in turn, release a number of proteins including platelet factor 4, RANTES, P-selectin, sCD40L and metaloproteinases, further escalating inflammation [10]. Macrophages infiltrate the vessel wall and secrete Interleukin-1β (IL-1β), Tumor Necrosis Factor-α, catepsins, and matrix metaloproteinases. They also phagocyte oxLDL depots and become ‘foam cells’. Foam cells further produce cytokines, growth factors, Interferon-γ and metaloproteinases. This causes an infiltration of the intima by vascular smooth muscle cells (VSMCs) [11]. VSMCs also change their phenotype and excrete mostly type I collagen, elastin and proteoglycans, enlarging the plaque volume [12]. This is followed by an accumulation of free, amorphic cholesterol crystals in the plaque and death of cells involved in the process. Cell death, through apoptosis or necrosis, is the basis for the formation and expansion of the necrotic core [13]. The expansion of the necrotic core leads to thinning of the fibrous cap covering the lesion, which makes it vulnerable to erosion or rupture and consequent thrombosis. Macrophages and mastocytes release metaloproteinases, which target collagen, further increasing the risk of rupture [14]. New, dysfunctional vessels sprout from vasa vasorum and infiltrate the plaque leading to local hemorrhages and causing further destabilization. Post-apoptotic debris and other components of the extracellular matrix within the plaque serve as foci of calcification, which can progress to a varying extent [15,16].

There is a vast number of cytokines implicated in the initiation and progression of atherosclerosis, but they are not the focus of this paper and have been reviewed elsewhere [17]. What deserves special attention in the context of mitochondrial involvement in atherosclerosis are reactive oxygen species and reactive nitrogen species (ROS and RNS, respectively). It has been shown that cells located in the plaque activated via pro-inflammatory signals (mainly Tumor Necrosis Factor-α, IL-1), angiotensin II or mechanical stimuli become a rich source of free radicals originating in the mitochondria [18]. ROS and RNS, in turn, further augment inflammatory responses and influence a number of cellular processes such as: adhesion, migration, proliferation and differentiation, all critical in the development of atherosclerosis [19,20]. There are several sources of oxidants such as myeloperoxidase, lipoxygenases, uncoupled endothelial nitric oxide synthase, however the mitochondrial NADPH oxidase is thought to be the predominant source of ROS and RNS [21,22].

The final stage of atherosclerosis—complications—is where apoptosis plays a special role. Plaque VSMCs present a relatively high apoptotic rate via c-myc, a proapoptotic protein [23]. Apoptosis of ECs and VSMCs enhance plaque vulnerability, hence increasing the risk of thromboembolic complications [24,25]. Additionally, macrophage apoptosis in mature plaques is involved in the further progression of the disease [26]. Apoptotic macrophages are abundant at sites of plaque rupture, as compared with stable lesions in material from patients who had undergone sudden cardiac death [27].

## 3. Profilin 1: General Information

Profilin family members (1–4) were first identified in the 1970s as actin-sequestering proteins [28]. Since then, a growing amount of data has suggested that proflin1, the most abundant human profilin isoform, is much more than just an actin regulator. For brevity, we will refer to Profilin-1 as ‘profilin’ throughout this review.

Profilin is a conservative actin-associated protein crucial for basic cell processes, such as proliferation, migration, cell-cell and cell-matrix interactions [29]. Murine embryos with a profilin gene knockout are unable to survive beyond a two-cell stage, which underscores its importance for the physiology of the cell [30]. Although actin regulation is its best-known function, profilin is much more than just this. It localizes not only in sub-membrane areas, where it interplays with actin, but also in the nucleus, within spliceosomes, Cajal bodies and gems, suggesting that it could be involved in post-transcription regulation of gene expression [31]. In support of the probable gen-regulating function, it was shown that profilin itself remains within the nucleus, while profilin-actin complexes are swiftly removed to the cytoplasm by exportin 6 [32].

*PFN1*, the gene encoding profilin, consists of three exones and is located in the short arm of chromosome 17 (17p13.2 subband) [33]. A microdeletion of this region has been implicated in the patophysiology of Miller-Dieker syndrome, a congenital disorder involving facial dysmorphia and lisencephaly [34]. Thus far, the role of *PFN 1* mutations in humans has been best characterized in the patophysiology of amyotrophic lateral sclerosis. It was shown that C71G, M114T, G118V and E117G mutations in *PFN1*, lead to structural and conformational alterations, which cause its aggregation and ubiquitination. This, in turn, impedes actin polymerization, decreases axonal growth and causes neuronal dysfunction [35,36]. It was also reported that gain-of-function mutations in *PFN1* might render a form of profilin, which causes conformational defects in TAR DNA-binding protein 43, a crucial molecule in the patophysiology of amyotrophic lateral sclerosis [37]. Moreover, post-translational profilin modifications (mostly phosphorylation) and changes in profilin expression have been implicated in a number of human malignancies, such as breast, pancreatic, renal and bladder cancer, as well as glioblastoma [38,39,40,41,42]. There have been some studies on profilin genetic variants in murine models showing that it is crucial for peripheral nervous system myelination [43] as well as motor neuron disease development [44].

Not much is known about the regulation of *PFN1* expression. However, there are reports showing that extracellular stimulation with oxysterol or angiotensin II activates *PFN1* expression via the JAK2/STAT3 pathway [45,46].

The structure of profilin determines its ability to bind three groups of partners: actin, phosphatidylinositol 4,5-bisphosphate (PIP2) and poly-proline containing proteins. One of the PIP2-binding regions is located within the actin-binding region, suggesting that these two molecules compete for the binding site [47,48,49]. Additionally to the three classes of molecular partners, profilin binds to microtubules via several surface residues and enhances their growth [50].

### 3.1. Profilin in Atherosclerosis

Profilin was found to be more abundant in coronary atherosclerotic plaques in comparison to healthy vessel wall fragments harvested from individuals with typical risk factors and severe coronary artery disease. Intriguingly, profilin was found in both ECs and in the extracellular matrix, suggesting that it may play a role in cell–cell communication. Indeed, profilin stimulation of rat and human VSMCs led to a 3–4-fold increase in DNA synthesis and migration rate. These effects were dose-dependent and relied on the classical growth factor-related pathways: PI3K/Akt and Ras/Raf/MEK/Src and Erk1/2. The authors also suggested that proatherogenic diet might enhance profilin expression [51]. Romeo et al. showed that Ldlr -/- mice lacking one copy of the profilin encoding (*PFN1*) gene were characterized by an antiatherogenic phenotype: higher endothelial nitric oxide synthase 3 activity, lower VCAM-1 expression, and lower oxLDL uptake by macrophages and lower inflammatory response to oxLDL as compared to Ldlr -/-. PFN1 -/- mice [52]. Dardik et al. reported that exposing human ECs to homocysteine under flow, but not static conditions led to an overexpression of profilin and underexpression of α-catenin, both of which are involved in actin cytoskeleton regulation. This was accompanied by a loss of α-catenin from intercellular junctions. These results suggest that homocysteine causes endothelial damage in dynamic conditions and that this occurs through cytoskeletal disruption mediated by high profilin levels [53]. Profilin may also exert pro-atherogenic effects through VSMCs. It was shown that stimulation with angiotensin II caused an up-regulation of the profilin gene expression with a subsequent proliferation of rat aortic VSMCs [46]. Since arterial hypertension is one of the main established causes of atherosclerosis, profilin may pose a molecular link between hypertension and plaque formation. 

### 3.2. Profilin in Coronary Artery Disease and Diabetes

Diabetes is one of the strongest factors influencing the progression of atherosclerosis and a predictor of poor percutaneous coronary intervention (PCI) outcomes [54]. Chronic hyperglycemia causes endothelial dysfunction, which is key for atherosclerosis development. Advanced Glycation End-Products (AGEs) are an early indicator of coronary artery disease in diabetic patients [55]. Li et al. showed that ECs exposed to AGEs overexpressed profilin, measured as mRNA and the concentration of the protein itself. This was associated with a cytoskeleton reorganization, a rise in ICAM-1, asymmetric dimethylarginine and a decrease in nitric oxide production. Blocking profilin ameliorated these effects. This shows that AGEs-related endothelial dysfunction is profilin-dependent [56]. Similar results were shown for diabetic cardiomyopathy, where profilin was upregulated in rat cardiomyocytes following chronic exposure to AGEs. The rise in profilin expression was associated with an increase in Rho, RAGE and p65. Again, silencing profilin ameliorated these changes [57]. The expression of the profilin gene in ECs was up-regulated by LDL cholesterol [58]. Furthermore, 7-ketocholesterol enhanced the transcription of *PFN1* gene in aortic endothelial cells in diabetic rats via JAK2/STAT3 activation and this was dependent on oxysterol-binding protein-1 [45]. We have recently reported a lower serum concentration of profilin in patients with coronary artery disease and co-existing diabetes, as compared with non-diabetics. The mechanism behind this phenomenon remains to be investigated. We hypothesize that carbonylation of profilin, a common post-translational modification in chronic hyperglycemia, may change the affinity of profilin to the antibody used in ELISA, leading to a lower reading [59].

### 3.3. Profilin in Myocardial Infarction

Hao et al. studied profilin expression in aortic ECs harvested from rats with myocardial infarction (MI) induced by left anterior descending artery ligation. They found that profilin and ERK1/2 were overexpressed in MI rats and correlated with the extent of myocardial damage as per troponin T and creatine kinase MB measurements. This was accompanied by a rise in the expression of mRNA of pro-apoptotic proteins such as p53, Fas, and Bax, whereas the mRNA for anti-apoptotic Bcl-2 expression was reduced. Moreover, ECs from MI-rats showed signs of dysfunction by higher endothelial microparticles production and lower nitric oxide release [60]. This suggests that the activation of ERK1/2 and profilin may be a mechanism of MI-related systemic endothelial dysfunction and apoptosis. Endothelial dysfunction, in turn, leads to a higher incidence of major adverse events and unfavorable myocardial remodeling in MI survivors [61]. Moreover, profilin is a significant component of thrombi retrieved from infarct-related arteries in ST-elevation myocardial infarction (STEMI) patients. Analysis of these thrombi showed that they have different compositions regarding actin cytoskeleton and associated proteins, including: β-actin, tropomyosin-3, -4 and profilin. Interestingly, profilin co-localized with platelets and leukocytes within the thrombus structure, suggesting its source. Profilin was more abundant in fresh thrombi and its levels decreased with time; with an opposite dynamic in the blood [62]. In our study, serum profilin concentrations were inversely proportionate to the onset-of-pain to reperfusion time in MI patients. Moreover, serum profilin was significantly lower in patients with an impaired post-intervention flow in the infarct-related artery, suggesting that it may be involved in coronary microvascular obstruction and ischemia/reperfusion injury [63]. The results from studies on profilin in atherosclerosis and coronary artery disease have been summarized in Table 1.

The mechanism in which profilin is implicated in atherosclerosis is unknown. Moreover, it is unclear whether its pronounced presence in atherosclerotic plaques is a source of the problem or its consequence. It is mostly postulated that profilin augments the development of plaques. However, one cannot exclude that its up-regulation is a result of the activation of cellular defense systems. Considering the fundamental role of profilin in regulating actin dynamics and the importance of the cytoskeleton in regulating a number of aspects in mitochondrial biology, the link between profilin and mitochondria is one exciting pathway to explore. Below, we discuss the ways in which mitochondria are involved in the pathogenesis of atherosclerosis and coronary artery disease with possible profilin involvement. Since the effects of profilin in the pathophysiology of atherosclerosis are ambiguous, we gathered the reported positive and negative effects of profilin in the pathophysiology of atherosclerosis in Figure 1. Mitochondria-related pathways are discussed below.

## 4. Mitochondria and Profilin in the Pathogenesis of Atherosclerosis

### 4.1. Mitochondrial Transport and Energy Production

The proper morphology and function of the mitochondria rely on the cytoskeleton. Mutation of actin-regulating proteins such as the Arp2/3 complex, cofilin or profilin led to a disorganization of actin cytoskeleton and distortion of mitochondrial morphology [66]. Furthermore, mitochondria need to be transported to regions of the cell, which depend on stable energy production. This transport relies on both, microtubule and actin-dependent transport. It was shown by Morris and Hollenbeck that pharmacological blocking of microtubules and leaving only actin available for transport led to a decrease in the velocity and distance traveled by mitochondria, with retrograde transport being favoured. When both actin and microtubules were inhibited, all mitochondrial mobility was blocked [67]. Therefore, actin cytoskeleton is important for a short-distance, mostly retrograde, mitochondrial movement. Once the mitochondria reach their destination, it is critically important that they remain anchored to regions of high energy-demand. This process relies largely on the proper function of the actin cytoskeleton. For example, in neurons, mitochondria are trafficked and anchored at sites stimulated by the Nerve Growth Factor in a TrkA-mediated fashion [68]. Since profilin is crucial for actin filament dynamics, it is probable that its dysfunction could impair mitochondrial movement inside the cell with negative implications for aerobic energy generation. Low aerobic capacity is known to worsen the exercise tolerance and outcomes, including mortality, of patients with coronary artery disease [69]. Hence, defective profilin function could impair aerobic respiration and negatively influence clinical outcomes in patients with coronary artery disease. This is a plausible mechanism, since dysfunctional profilin variants were reported in simple Eukaryotes [70], as well as in human pathology, such as amyotrophic lateral sclerosis (see Section 3).

### 4.2. Fusion and Fission

Mitochondria are dynamic organelles that undergo constant fusion and fission [71]. A proper balance between these processes is crucial, as it serves to distribute proteins and metabolites across the whole compartment, as well as to minimize oxidative damage [72]. Mitochondria with an excess of free radicals fuse with others in order to “dilute” ROS and RNS in a larger organelle. They may also be isolated by fission and directed towards mitophagy. Finally, in the case of overwhelming damage, the cell may undergo apoptosis [73].

Inverted formin 2 (INF2) is a protein that connects mitochondrial fission to the actin cytoskeleton. Endoplasmic reticulum-bound INF2 forms actin filaments in the proximity of mitochondria, which is followed by Dynamin-1-like protein (Drp1) mediated mitochondrial fission [74]. Profilin interacts directly with INF2 and accelerates the ADP to ATP exchange in the process of actin polymerization [75]. Therefore, the direct profilin-INF2 interactions most likely facilitate mitochondrial fission via Drp1. An experimental disruption of this process by a down-regulation of Drp1 led to a reduction in mitochondrial fission and consequently to less ROS generation, better endothelial function and a reduction in atherosclerotic plaques in streptozotocin-induced diabetic ApoE -/-; mice [76]. Perhaps excessive profilin expression within the atherosclerotic plaque destroys the mitochondrial fission/fusion balance in favor of fission, leading to exacerbation of the disease. This could be one way to explain the abundance of profilin within atherosclerotic plaques.

### 4.3. Apoptosis

As mentioned before, apoptosis is one of the key processes involved in the progression and complications of atherosclerosis. BCL-2, a family of mitochondrial proteins, play crucial roles both in promoting (BAX, BAK, and BOK) and blocking (BCL-2, BCL-XL, BCL-W, MCL-1, and BFL-1/A1) apoptosis [77]. There is a growing amount of evidence pointing to the involvement of the actin cytoskeleton and associated proteins in mitochondria-mediated apoptosis. It was demonstrated that an accumulation of actin in the proximity of mitochondria directly precedes the hallmarks of apoptosis: the release of cytochrome c, BAX translocation to the mitochondria, mitochondrial fission and condensation of nuclear DNA. It is hypothesized that actin accumulation in the proximity of mitochondria facilitates the transport of pro-apoptotic proteins (BAX) from the cytosol to the mitochondrion [78]. Moreover, a pharmacological stabilization of the actin cytoskeleton by jasplakinolide and its destabilization by cytochalasin D were both shown to exert proapoptotic effects. Caspase-3, a mitochondrial initiator of apoptosis was elevated in both cases [79,80]. Mitochondria-induced apoptosis may also be affected by actin-associated proteins. Gelsolin, an actin assembly/disassembly regulator, closes the voltage-dependent anion channel and maintains the physiological mitochondrial membrane potential. Therefore, it reduces the outflow of cytochrome c to the cytoplasm and blocks apoptosis [81]. However, cofilin—a partner of profilin in regulating actin depolymerization—was demonstrated to have pro-apoptotic properties. The translocation of dephosphorylated cofilin and Drp1 to the mitochondria enhanced mitochondrial fission and apoptosis [82]. There is growing evidence that profilin itself is a directly pro-apoptotic protein and that it exerts this function via mitochondrial pathways. Yao et al. reported that profilin mediated the up-regulation, activation and mitochondrial translocation of p53. This induced apoptosis by mitochondrial, transcription-independent pathways [83]. Furthermore, Zaidi and associates investigated the role of profilin in the survival of breast cancer cells. They transfected a breast cancer cell line with the profilin gene using a high-level constitutive expression vector (pcDNA 3.1). This led to the engineering of “profilin stable” cells that were then compared with wild type (unmodified) cells after treatment with apoptosis inducers: doxorubicin, oleandrin, paclitaxel or vinblastine. They showed a 70–90% death rate in “profilin stable” cells as compared with 40–50% of unmodified cells. Profilin exerted its pro-apoptotic properties by decreasing the binding capacity and activity of NF-κB, as well as through the phosphorylation of p53. The phosphorylated p53 was translocated to the mitochondrion, where it triggered caspase 9-dependent apoptotic processes. In this way, profilin may increase mitochondria-driven apoptosis [84]. In support of these results, several inhibitors of cellular proliferation and migration were shown to up-regulate profilin [85,86]. On the other hand, in murine hematopoietic stem cells a knock-out of profilin gene led to apoptosis and an increase in ROS production via mitochondrial respiration. These effects happened due to the downstream activation of early growth response protein 1 (EGR1), a transcription factor involved fibrinogenesis and fibrotic processes in response to extracellular stimuli. This suggests that profilin may regulate mitochondrial function and apoptosis through gene expression [87]. Interestingly, EGR1 was implicated in atherosclerosis and activating the post-ischemia inflammatory response [88,89]. It is possible that the final effect that profilin may have on mitochondria-driven apoptosis depends on additional regulators or tissue type and requires further research. Our own research shows low profilin levels in MI patients with suboptimal infarct-related artery flow post-PCI—a symptom of inadequate tissue perfusion in the infarct area [63]. This could be due to a profilin-EGR1-mitochondria driven inflammation in the infarct region.

### 4.4. Profilin-SIRT3 Interactions

The NAD-dependent deacetylase sirtuin-3 (SIRT3) is a major mitochondrial NAD-deacetylase, which promotes the Krebs cycle and aerobic ATP production via the mitochondrial electron transport chain, maintains proper potential across the mitochondrial membrane and reduces ROS levels. It is also implicated in a number of metabolic effects, cell survival, inflammation and cellular reaction to stress [90]. SIRT3 seems to have several protective roles in the pathogenesis of atherosclerosis. Polymorphisms in SIRT genes were linked to the advancement of atherosclerotic plaques, which was attributed to ROS imbalance and ROS-related damage [91]. SIRT3 inhibits trimethylamine-N-oxide (TMAO), a potent activator of the NLRP3 inflammasome, and caspase-1—a proinflammatory and proapoptotic enzyme, which yields IL-1β and interleukin-18 [92]. Also, Sirt-3 deficient mice on a high-cholesterol diet developed endothelial dysfunction that was dependent on low superoxide dismutase 2 activity (SOD2) and high ROS generation [93]. SIRT3 was also shown to reduce ROS-production and prevent apoptosis by deacetylating Ku70 and it’s binding with Bax. This prevented Bax from translocating to the mitochondrion and setting off an Apoptosis Inducing Factor (AIF)-dependent cascade. Ergo, SIRT3 prevents stress-induced cell death [94,95]. Taking into consideration the importance of apoptosis in thromboembolic complications of atherosclerosis, SIRT3 may be one of the most important proteins protecting against MI and stroke.

SIRT3 also has important metabolic functions with implications for atherosclerosis. SIRT3-knockout mice placed on a high-fat diet show insufficient fatty acid β-oxidation and are prone to metabolic syndrome, including insulin resistance and hepatosteatosis. The case is similar in humans: a single nucleotide polymorphism in the human *SIRT3* gene reduced the enzyme activity leading to mitochondrial protein acetylation and metabolic syndrome [96].

A potentially game-changing result was reported by Yao et al. They showed that profilin interacts with SIRT3 directly and specifically, both in vitro and in vivo. A transfection with profilin resulted in a rise in SIRT3 expression. Moreover, the profilin-SIRT3 interaction led to the down-regulation of hypoxia-inducible factor 1α (HIF-1α) [40]. HIF-1α is a transcription factor for a number of genes involved in cell proliferation, angiogenesis (including Vascular Endothelial Growth Factor) and glucose metabolism [97]. Importantly, HIF-1α is highly expressed in smooth muscle cells and cardiomyocytes [98], which underscore the potential role of a possible profilin-SIRT3-HIF-1α axis in atherosclerosis. It is plausible that profilin may influence SIRT3-mediated processes involved in the formation and progression of atherosclerotic plaques (ROS scavenging, inflammation, and apoptosis).

SIRT3 is also protective against hypertension, a prominent risk factor for coronary artery disease. Dikalova et al. demonstrated a SIRT3 deficiency with a SOD2 acetylation (inactivation) in the blood mononuclear cell fraction of hypertensive individuals. SOD2 expression itself was not influenced. This suggests that SIRT3 deficiency leads to SOD2 hyperacetylation and elevated ROS, which contribute to hypertension [99]. However, profilin was shown to promote hypertension and vascular remodeling in rodents. This was partially also mediated via an increased level of free radicals (peroxynitrite), suggesting another way in which SIRT3-profilin interplay could influence atherosclerosis [64,100].

### 4.5. Ischemia/Reperfusion Injury

Yang at al. showed that cardiomyocytes exposed to stress by nutrient withdrawal managed to maintain physiological NAD+ levels, which is essential for cell survival. This was achieved due to a compensatory rise in nicotinamide phosphoribosyltransferase (Nampt) expression, as well as functional mitochondrial SIRT3 and SIRT4 [101]. This may be critically important in acute myocardial ischemia, where due to artery blockage, oxygen and nutrients are lacking in the myocardium. The proper function of SIRT3 could maintain cell viability and integrity up to the time of reperfusion and restoration of nutrient flow. In this case, a dysfunction of SIRT3 could increase the risk of ischemia-reperfusion injury. In fact, hearts harvested from SIRT3-knockout mice, which underwent myocardial infarction, showed more pronounced ischemia/reperfusion injury in comparison with wild type specimens. This was accompanied by a larger extent of mitochondrial permeability transition pore opening and ROS production [102]. It is possible that profilin facilitates the proper action of SIRT3 and prevents ischemia-reperfusion injury. This notion is supported by our own results. As mentioned before, we recorded lower profilin concentrations in MI-patients with inadequate post-PCI coronary flow, which is an angiographical sign of microvascular obstruction and ischemia/reperfusion injury [63]. The observed low serum profilin levels do not prove a causal role of profilin in ischemia-reperfusion injury; however, in light of the role of SIRT3 and its direct interaction with profilin, this may constitute a promising research area.

Excessive mitochondrial fission was also reported to cause ischemia/reperfusion injury [103]. This is another mechanism where profilin (acting via INF2) could exacerbate ischemia/reperfusion injury. The possible involvement of profilin in mitochondrial fission has been discussed above. A putative model for profilin-mitochondria associations in processes involved in the pathogenesis of atherosclerosis is presented in Figure 2.

## 5. Knowledge Gap

Very little is known about the mechanisms in which profilin acts in atherosclerosis on different stages of the disease, which opens a completely novel area of research. One exciting field to explore is the involvement of profilin with mitochondria. The direct interaction between profilin and mitochondrial deacetylase SIRT3 seems to be particularly promising [40]. Further investigation into the nature of this interaction is needed, especially regarding final SIRT3 effects relevant in atherosclerosis, such as: ROS generation, NLRP3-mediated inflammation, apoptosis and cellular reaction to stress. There are data suggesting that profilin increases the expression of SIRT3 [84]; however, it is unclear whether they act in a synergistic manner. Moreover, a growing amount of data point to a possible profilin and mitochondrial involvement in other cardiovascular pathologies, such as cardiac hypertrophy and hypertension. Further research into the potential link between profilin and mitochondria in heart-related issues could change our understanding of atherosclerosis and other heart-related conditions.

## 6. Conclusions

A growing amount of data associate profilin 1, a protein crucial for cell biology, with the pathogenesis of coronary artery disease. It is increasingly clear that profilin functions far beyond its well-known role as an actin dynamic regulator. The analysis of the network of profilin’s molecular partners and interactions allows us to hypothesize that there are several areas in which profilin may exert pro or antiatherogenic effects via mitochondria. These include: regulation of aerobic energy generation, mitochondrial fission, apoptosis, ROS generation and neutralization. The direct interaction between profilin and SIRT-3 seems to be particularly promising in shedding new light on the pathogenesis of atherosclerosis.

## Figures and Tables

**Figure 1 ijms-22-01100-f001:**
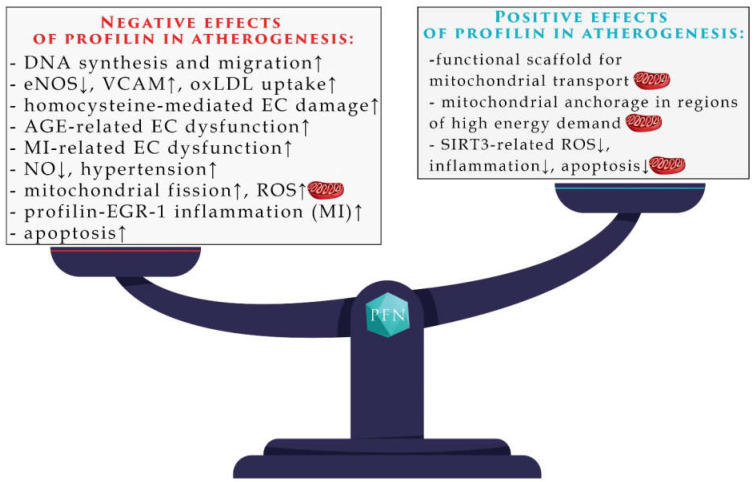
A summary of profilin involvement in the pathogenesis of atherosclerosis. Details in text. AGE: Advanced Glycation End-Products; eNOS: endothelial nitric oxide synthase; EC: Endothelial Cells; MI: myocardial infarction; oxLDL: oxidized low-density lipoprotein; PFN: profilin; ROS: Reactive Oxygen Species; EGR: Early Growth Response Protein.

**Figure 2 ijms-22-01100-f002:**
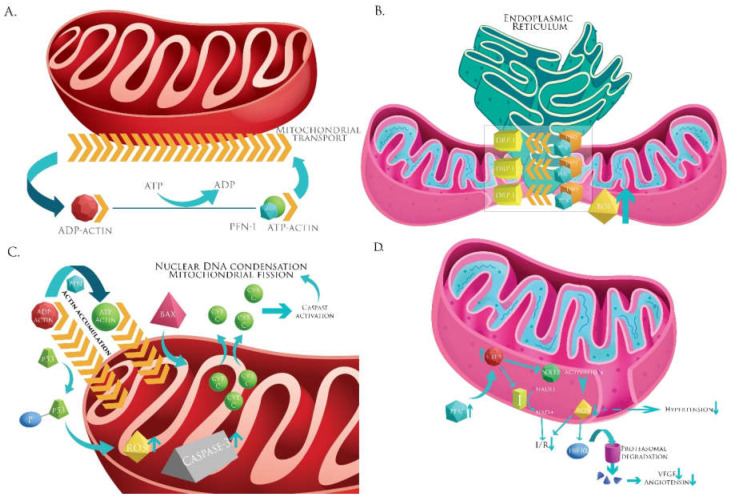
A putative model for profilin-mitochondria associations in processes involved in the pathogenesis of atherosclerosis. (**A**) Mitochondrial transport. Profilin accelerates the rate of actin polimerization by ADP-actin to ATP-actin conversion, building a scaffold for mitochondrial transport. (**B**) Mitochondrial fission. Profilin interacts with Endoplasmic reticulum-bound INF2 and accelerates the ADP to ATP exchange in actin polymerization. This provides a platform for Drp1-mediated mitochondrial fission. (**C**) Apoptosis. Profilin-mediated actin accumulation leads to translocation of Bax to the mitochondria with subsequent cytochrome c release. The level of caspase-3 rises. Profilin phosphorylates p53, which translocates to the mitochondria activating other apoptotic pathways. ROS levels increase. There are followed by nuclear DNA condensation and enhanced mitochondrial fission. (**D**) Profilin-SIRT-3 interactions. Profilin increases SIRT-3 levels. SIRT-3 deacetylates (activates) SOD-2, leading to a decrease in ROS. A low ROS level promotes the degradation of HIF-1α and prevents VEGF transription and local neoangiogenesis. SIRT-3 also increases NAD+ levels, together with decreased ROS protect against Ischemia/Reperfusion Injury. Abbreviations: ADP: Adenosine diphosphate; ATP: Adenosine triphosphate; I: complex I of the electron transport chain; cyt c: cytochorme c; Drp1: Dynamin-1-like protein; HIF-1α: Hypoxia Inducible Factor-1α; INF-2: Inverted Formin-2; I/R: ischemia/reperfusion injury; NAD+/NADH: Nicotinamide Adenine Dinucleotide redox pair; PFN-1: Profilin 1; ROS: Reactive Oxygen Species; SIRT-3: NAD-dependent Deacetylase Sirtuin-3; SOD-2: Superoxide Dismutase 2; VEGF: Vascular Endothelial Growth Factor.

**Table 1 ijms-22-01100-t001:** Summary of the main conclusions from studies on profilin in atherosclerosis and coronary artery disease. LDL: low density lipoprotein; PCI: percutaneous coronary intervention; STEMI: ST-segment elevation myocardial infarction.

Publication Year	Authors	Main Conclusion	Reference
2004	Romeo G et al.	Profilin is abundant in aortic atherosclerotic plaques and leads to LDL-dependent endothelial dysfunction in hyperglycemia	[58]
2007	Moustafa-Bayoumi et al.	Profilin overexpression leads to vascular hypertrophy and hypertension	[64]
2010	Caglayan et al.	Profilin is expressed in atherosclerotic plaques and exerts proatherogenic effects in vascular smooth muscle cells	[51]
2013	Li Z et al.	Profilin exerts proatherogenic effects in response to chronic hyperglyceamia	[56]
2013	Zhao et al.	Profilin 1 contributes to cardiac hypertrophy and interferes with nitric oxide production in hypertensive rats	[65]
2015	Ramaiola et al.	Profilin is present in thrombi from infarct-related arteries in STEMI patients. The amount of profilin decreases with thrombus “age”	[62]
2017	Yang et al.	Profilin is up-regulated in cardiomyocytes in response to chronic exposure to advanced glycation end-products and contributes to cardiotoxicity	[57]
2017	Hao et al.	Profilin is overexpressed in aortic endothelial cells during myocardial infarction	[60]
2019	Paszek et al.	Serum profilin levels in coronary artery disease are associated with diabetes, family history and multivessel disease	[59]
2020	Paszek et al.	Profilin serum concentration depends on the time of symptom duration, post-PCI flow in the infarct-related artery and P2Y12 administration in myocardial infarction patients	[63]

## Data Availability

Not applicable.

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
