# Peer review of "Profilin 1 and Mitochondria—Partners in the Pathogenesis of Coronary Artery Disease?"

_ijms, 2021, doi:10.3390/ijms22031100_

Round 1
Reviewer 1 Report
This is a well written article reviewing the role of profilin in atherosclerosis and associated pathologies. Paszek et al., summarized the current state of knowledge for possible molecular pathways through which profilin may act/affect mitochondrial dynamics.
Comments:
Line 34: Replace ‘male sex’ with ‘gender’
Line 55: Replace ‘gene expression lead to a’ with ‘gene expression leading to a’
Profilin 1 – general information:
Is there any information on known profilin mutations in humans and it’s implications?
Add information on models of profilin knockout/mutations in higher animals, if any.
It would be helpful for the readers if the authors can also add details of the timing, level and regulation of profilin gene expression.
Line 120: ‘led to a ca. 3-4-fold increase..’ What is ca.?
Line 137-139; line 155-156: Insert appropriate reference citation at the end of respective sentences.
Line 193: Replace ‘profilin led to a..’ with ‘profilin lead to a..’
Line 208: “Hence, defective profilin function could impair aerobic respiration..” Is there any evidence in the literature about ‘defective profilin’ or is it just a speculation?
Line 252: ‘“profilin stable”, and wild type (unmodified)..’ Give more details of the cell line and explain the mechanistic difference between the two.
Overall, the review reports both pro- or antiatherogenic effects of profilin, which is confusing. For the benefit of the readers, I’d recommend adding a section which clearly outlines evidence of pro and antiatherogenic effects of profilin. The authors may avoid adding extra text by creating an illustrative diagram to explain pro- and antiatherogenic effects.
Author Response
Reviewer 1
This is a well written article reviewing the role of profilin in atherosclerosis and associated pathologies. Paszek et al., summarized the current state of knowledge for possible molecular pathways through which profilin may act/affect mitochondrial dynamics.
Thank you for your interest in our work.
Comments:
Line 34: Replace ‘male sex’ with ‘gender’
Line 55: Replace ‘gene expression lead to a’ with ‘gene expression leading to a’
Thank you for your comments. We have updated the text appropriately. We have corrected the verb in the past tense.
Profilin 1 – general information:
Is there any information on known profilin mutations in humans and it’s implications?
We have included information on the role of the PFN gene – containing region of chromosome 17 deletion in Miller – Dieker syndrome, as well as PFN mutations in amyotrophic lateral sclerosis and also pointed out the role of profilin post-translational modifications and expression changes in some human cancers.
Add information on models of profilin knockout/mutations in higher animals, if any.
We had previously mentioned examples of profilin knockouts in rodents: a mice model, where a double profilin knockout embro fails to survive beyond the two-cell stage of development (par. 3 – Profilin – general information), the role of profilin knockout in Ldl-r -/- mice (par. 3.1 Profilin in atherosclerosis) and on the effects of profilin knockout in murine hematopoietic stem cells (par. 4.3 Apoptosis). Upon your remark, we have additionally included two other examples of murine profilin gene knock-out models (par. 3.1 Profilin 1 – general information. We were unable to find genetic models directed at profilin in higher animals other than rodents.
It would be helpful for the readers if the authors can also add details of the timing, level and regulation of profilin gene expression.
We added some information on the PFN gene expression in par. 3 Profilin – general information, Beyond this, not much is known about the constitutive expression of the profilin gene.
Line 120: ‘led to a ca. 3-4-fold increase..’ What is ca.?
Ca. was meant as ‘around’. We erased it as it might have been confusing and did not change the meaning of the sentence.
Line 137-139; line 155-156: Insert appropriate reference citation at the end of respective sentences.
Line 193: Replace ‘profilin led to a..’ with ‘profilin lead to a..’
Thank you for your comments. We updated the text accordingly.
Line 208: “Hence, defective profilin function could impair aerobic respiration..” Is there any evidence in the literature about ‘defective profilin’ or is it just a speculation?
This has been described in relation to amyotrophic lateral sclerosis, where structural changes cause profilin aggregation and its inability to perform its function and we have added this information (see par. 3. Profilin 1 – general information). There have been reports on defects in profilin function in yeast and, upon your comment, we have also cited this research in this part of discussion on possible profilin defective function.
Line 252: ‘“profilin stable”, and wild type (unmodified)..’ Give more details of the cell line and explain the mechanistic difference between the two.
We have added this information in par. 4.2.
Overall, the review reports both pro- or antiatherogenic effects of profilin, which is confusing. For the benefit of the readers, I’d recommend adding a section which clearly outlines evidence of pro and antiatherogenic effects of profilin. The authors may avoid adding extra text by creating an illustrative diagram to explain pro- and antiatherogenic effects.
Thank you for your comment. We added a new figure according to Reviewer’s recommendation.
Reviewer 2 Report
Atherosclerosis is a multifactorial arterial disease constituting the pathological basis of diverse intractable cardiovascular diseases, including myocardial infarction, heart failure, stroke and so on. Therefore, its molecular mechanism has gained considerable attention. Beyond the concept of inflammation, Paszek E et al. summarized the latest findings among profilin 1, mitochondrial dynamics, and different heart diseases, providing a different angle of perspective. However, the relationship between Pfn1 and atherosclerosis has not been fully discussed. There are several issues need to be further addressed before consideration for the publication.
- The main structure and logic of the current manuscript need to be further improved, avoiding simply accumulating information. Instead of discussing diseases like IR, MI, diabetes, etc., authors should focus on the potential role of Pfn1 in every aspect of atherosclerosis. Or how to apply the knowledge of Pfn1 from other diseases into atherosclerosis.
- The molecular basis of Pfn1 at different stages of atherosclerosis and the role of Pfn1 in different cell types within plaques need more attention.
- When introducing the generation of free radicals (Page 2, Line 80 – 81), the significance of mitochondrial origin has not been fully emphasized. In other words, why authors focused on the mitochondria-originated ROS/RNS, not other sources?
- Authors should apply the updated and correct information for Pfn1 (Page 3, Line 98). Pfn1 is ubiquitously expressed throughout the mammal body, including muscle cells. Mutation of Pfn1 causes muscle atrophy and dysfunction (https://doi.org/10.1093/hmg/ddw429).
- Although Pfn1 was reported to interact with SIRT3, that doesn’t mean that Pfn1 has the same role as SIRT3 even if they showed a similar expression trend (Part 4.5).
- The biomarker or a risk factor doesn’t mean that factor actually mediates the pathological process (Page 9, Line 322- 324).
Minor:
- Instead of using “isoforms” (Page 3, Line 95), it should be defined as a profilin family member or paralog for Pfn1 – 4, which are encoded by distinct genes.
- Authors should provide ref for main points (Page 8, Line 293 - 295).
Author Response
Reviewer 2
Atherosclerosis is a multifactorial arterial disease constituting the pathological basis of diverse intractable cardiovascular diseases, including myocardial infarction, heart failure, stroke and so on. Therefore, its molecular mechanism has gained considerable attention. Beyond the concept of inflammation, Paszek E et al. summarized the latest findings among profilin 1, mitochondrial dynamics, and different heart diseases, providing a different angle of perspective. However, the relationship between Pfn1 and atherosclerosis has not been fully discussed. There are several issues need to be further addressed before consideration for the publication.
Thank you for your interest in our work.
- The main structure and logic of the current manuscript need to be further improved, avoiding simply accumulating information. Instead of discussing diseases like IR, MI, diabetes, etc., authors should focus on the potential role of Pfn1 in every aspect of atherosclerosis. Or how to apply the knowledge of Pfn1 from other diseases into atherosclerosis.
We feel this is one of two good ways to describe this topic. The suggested layout based on consecutive aspects of atherosclerosis is very logical and many reviews before have been successfully constructed this way. However, we feel that the current layout of our manuscript concentrating on clinical aspects of atherosclerosis may be more straightforward for clinicians who deal with complications like MI, I/R, diabetes-related complications etc on an everyday basis.
- The molecular basis of Pfn1 at different stages of atherosclerosis and the role of Pfn1 in different cell types within plaques need more attention.
We have added additional information about profilin in the proliferation of SMVCs. (see: par. 3.1). Nevertheless, to our knowledge, profilin involvement in atherosclerosis has been described in ECs and VSMCs only. We have attempted to summarize these. Not much is known on the exact molecular mechanisms of profilin action in atherosclerosis – further research in the area is needed. Hence, we suggest in this review profilin may act via mitochondrial pathways but this concept requires further research.
- When introducing the generation of free radicals (Page 2, Line 80 – 81), the significance of mitochondrial origin has not been fully emphasized. In other words, why authors focused on the mitochondria-originated ROS/RNS, not other sources?
There are multiple sources of ROS and RNS in the cells. However, since the review focuses on the possible interplay between profilin and mitochondria, we focused solely on this source of free radicals. We did not want to list all other sources of ROS, as they are not the topic of this review. Upon your remark however, we decided to add some information on other sources of ROS (par. 2)
- Authors should apply the updated and correct information for Pfn1 (Page 3, Line 98). Pfn1 is ubiquitously expressed throughout the mammal body, including muscle cells. Mutation of Pfn1 causes muscle atrophy and dysfunction (https://doi.org/10.1093/hmg/ddw429).
Thank you for this remark. Paper cited by the Reviewer tackles an issue of muscular dystrophy secondary to changes in neuron cells. Indeed, in that experiment it was possible to induce pfn expression in skeletal muscle, but this is not entirely a physiological situation, but rather an artificial construct.
- Although Pfn1 was reported to interact with SIRT3, that doesn’t mean that Pfn1 has the same role as SIRT3 even if they showed a similar expression trend (Part 4.5).
We agree, and we did not mean to imply otherwise. To underscore this, we have added a comment on this in the „knowledge gap” section.
- The biomarker or a risk factor doesn’t mean that factor actually mediates the pathological process (Page 9, Line 322- 324).
We agree and we have included a sentence to clarify this (see par. 4.5)
Minor:
- Instead of using “isoforms” (Page 3, Line 95), it should be defined as a profilin family member or paralog for Pfn1 – 4, which are encoded by distinct genes.
Thank you for your comment. This has been corrected.
- Authors should provide ref for main points (Page 8, Line 293 - 295).
Thank you for your comment. The references has been added.
Reviewer 3 Report
- The review is interesting and original. It provides new insights in the pathogenesis of endothelial dysfunction and atherosclerosis.
- The results are appropriately and significant, the conclusions are justified and supported by the results
- The review is well written, the data and analyses are very wall presented
- The conclusions are interesting , I consider that the paper will attract a wide readership
- The paper provides an advance towards the current knowledge
- The English language is appropriate and understandable?
Author Response
Thank you for your interest in our work
Round 2
Reviewer 2 Report
The authors clarified several points raised in the last review.